# Oncolytic Adenovirus, a New Treatment Strategy for Prostate Cancer

**DOI:** 10.3390/biomedicines10123262

**Published:** 2022-12-15

**Authors:** Kaiyi Yang, Shenghui Feng, Zhijun Luo

**Affiliations:** 1Department of Urology, Xiangya Hospital, Central South University, Changsha 410008, China; 2Provincial Key Laboratory of Tumour Pathogens and Molecular Pathology, Queen Mary School, Nanchang University, Nanchang 330031, China

**Keywords:** oncolytic virus, human adenovirus, prostate cancer, immunotherapy, gene therapy

## Abstract

Prostate cancer is the most common cancer and one of the leading causes of cancer mortality in males. Androgen-deprivation therapy (ADT) is an effective strategy to inhibit tumour growth at early stages. However, 10~50% of cases are estimated to progress to metastatic castration-resistant prostate cancer (mCRPC) which currently lacks effective treatments. Clinically, salvage treatment measures, such as endocrine therapy and chemotherapy, are mostly used for advanced prostate cancer, but their clinical outcomes are not ideal. When the existing clinical therapeutic methods can no longer inhibit the development of advanced prostate cancer, human adenovirus (HAdV)-based gene therapy and viral therapy present promising effects. Pre-clinical studies have shown its powerful oncolytic effect, and clinical studies are ongoing to further verify its effect and safety in prostate cancer treatment. Targeting the prostate by HAdV alone or in combination with radiotherapy and chemotherapy sheds light on patients with castration-resistant and advanced prostate cancer. This review summarizes the advantages of oncolytic virus-mediated cancer therapy, strategies of HAdV modification, and existing preclinical and clinical investigations of HAdV-mediated gene therapy to further evaluate the potential of oncolytic adenovirus in prostate cancer treatment.

## 1. Introduction

Oncolytic virus therapy has emerged as a promising tumour therapeutic method in recent years. By selecting weak pathogenic virus strains in nature or targeted genetic modifications of certain viruses, the construction of oncolytic viruses with the function of targeting and killing tumour cells becomes feasible in practice [1]. Compared with traditional tumour therapeutic strategies such as chemotherapy and radiotherapy, oncolytic viruses are unique in terms of action mechanism, specificity, and cross resistance [2]. Oncolytic virus therapy is characterized by high replication efficiency, salient killing effect, lower toxicity, and fewer side effects, which has become a new hotspot in the research of cancer treatment [3].

Currently, plenty of viruses have been used for oncolytic therapy, including adenovirus, herpes simplex virus type I (HSV-1) [4], Newcastle disease virus [5], measles virus [6], mumps virus [7], vaccinia virus [8], and vesicular stomatitis virus [9]. Two oncolytic viruses H101 and T-Vec have been approved for the treatment of cancer. Among them, H101 is a genetically modified oncolytic adenovirus. It was approved for marketing by the State Food and Drug Administration of China in 2005. It can be used for local injection in combination with chemotherapy to treat head and neck cancer, mainly nasopharyngeal cancer [10,11]. In 2015, the FDA approved type I herpes simplex modified virus T-Vec expressing granulocyte-macrophage colony-stimulating factor (GM-CSF) for the treatment of malignant melanoma [12].

The combination of oncolytic virus therapy and traditional chemotherapy, radiotherapy, ionizing radiation, and other treatment methods will be one direction for anticancer therapeutics in future [11,13]. In recent years, with the development of interdisciplinary approaches, the research on oncolytic viruses in the treatment of tumour has also entered a new era, and an increasing number of genetically modified oncolytic viruses have entered clinical trials [14,15,16,17,18,19,20,21,22].

Prostate cancer exhibits the second-highest mortality among all malignant tumours [23], associated with up to 4% (375,000/year) of cancer-related deaths globally [24]. Radical treatment is used for early localized prostate cancer, such as radical surgery, radiotherapy, brachytherapy, or high-energy ultrasound focusing. It is one of the most effective ways to treat localized cancer with radical prostatectomy. However, approximately 35% of patients with early-stage localized cancer and 50% of locally advanced prostate cancer recurs and metastasizes [25]. The 5-year survival rate of metastatic prostate cancer is 28% [26]. Most patients with metastatic cancer receive salvage therapy such as endocrine therapy and chemotherapy to relieve symptoms and prolong survival time. Androgen deprivation therapy can effectively inhibit the growth of prostate cancer at early stages, but castration resistance is estimated to develop in 10~50% of cases within 3 years of diagnosis [27,28,29]. Docetaxel-based quick therapy can increase the survival of patients with castration-resistant prostate cancer (CRPC), but failure eventually occurs [28,30]. Ra-223 is a radiopharmaceutical that improves overall survival (OS) in CRPC with bone metastases [31]. Later, another radiopharmaceutical, 177Lu-PSMA, was approved for the treatment of mCRPC patients after taxane-based chemotherapy [32]. However, their specificity and sensitivity are limited.

In recent years, with the elucidation of the mechanism underlying prostate cancer castration resistance and the androgen receptor signalling, drugs such as abiraterone and the new non-steroidal anti-androgen drug enzalutamide have been developed to inhibit the synthesis of intracellular androgen [33,34]. However, these drugs only delay tumour progression to a certain degree, and resistance will eventually be developed. In order to improve the therapeutic effects on prostate cancer and the outcomes of patients, new treatments including oncolytic adenovirus therapy are continually being explored [35].

## 2. Characteristics of Oncolytic Adenoviruses

Adenovirus is a DNA virus widely distributed in nature, where many types exist. Among them, the category infecting humans is generally called human adenovirus (HAdV).

### 2.1. Structure, Proteins, Genome, and Classification of HAdV

Adenovirus has no envelope and is about 70–90 nm in diameter [36,37]. It consists of a characteristic icosahedron that makes up its viral capsid encompassing double-stranded DNA and associated nucleoproteins to form the core. The protein capsid consists of three major proteins (hexon, penton, and fibre) and a small number of other proteins (Figure 1). Hexons constitute the majority of faces and edges of the icosahedron. There are 12 pentons located at the 12 vertices of the icosahedron. The size and number of fibrous knobs are species-specific and associated with the process of adhesion and binding of the virus to the receptor cells.

More than 57 serotypes of HAdV are divided into seven different subclasses/species A–G, causing various symptoms in humans upon infection. For example, subclasses B1, C, and E induce changes in respiratory systems due to high affinity, subclass B2 affects kidney and urinary tract, subclass F causes gastroenteritis, and some serotypes of subclass D lead to epidemic keratoconjunctivitis [38]. The most widely used oncolytic adenoviruses are serotypes 2 and 5 of subclass C (HAdVC2 and HAdVC5). They generally cause mild respiratory distress and are not tumorigenic.

HAdV shares a receptor with the Coxsackie virus called the Coxsackie and Adenovirus Receptor (CAR). Except for the receptor of subgroup B, which is CD46, other subgroups bind CAR (Table 1) [39].

The genome (including coding and non-coding regions) is about 26–46 Kb of dsDNA and encodes 23–46 proteins depending on the serotype. The coding region contains five early transcription units (E1A, E1B, E2, E3, and E4) encoding different regulatory factors, two delayed transcription units (IX and IVa2), and five late transcription units (L1–L5) encoding the viral structural protein (Figure 2) [41,42]. Among the early genes, E1A and E1B possess the ability of self-replication of the virus, and E3 is involved in the immune escape of the host.

### 2.2. HAdV Infection

As depicted in Figure 3, HAdV infects cells through the following major steps [43,44]: Firstly, the fibre knobs of HAdV bind to specific receptors on the host cell membrane, among which CAR is the most important receptor. After HAdV adhesion, the arginine-glycine-aspartic acid (RGD) sequence on the penton protein binds to integrin αvβ3 and αvβ5 on the cell surface. Secondly, HAdV is internalized into cells by endocytosis in clathrin-coated pits and the endosome is then fused with the lysosome. Thirdly, in the acidic environment of the lysosome, the conformation of the capsid changes followed by partial degradation of the virus particles, release from the lysosome and transport to microtubules. Finally, the released virus is moved along microtubules to the nucleus, where viral DNA enters the nucleus through the nuclear pore to proceed through the process of DNA replication and transcription of viral proteins.

### 2.3. Adenovirus Causes an Immune Response When It Enters Cells

Upon infection of host cells, the virus stimulates the release of interferon (IFN) and activation of Toll-like receptors (TLRs), which then triggers the innate immunity and adaptive immunity, leading to production of proinflammatory cytokines and chemokines (Figure 4) [45,46,47]. As a result, the immune system continuously regulates the infection process and thereby induces cell apoptosis or necrosis, which generates preventive or antiviral effects. The innate immune response directly limits virus replication and assembly and induces adaptive immunity [48].

### 2.4. Immune Escape of HAdV

The expression of multiple HAdV genes interferes with host immunity: (1) The E1A protein is an early virus transactivator. The E1A gene is transcribed and alternatively spliced soon after the viral DNA genome enters the nucleus. It is expressed in the early phase of infection and regulates cell cycle progression, cell apoptosis, immune evasion, tumorigenesis, and viral gene expression [49]. Cell cycle arrest is favourable to virus replication and required for the effective expression of other viral genes. E1A also inhibits NF-kB-dependent transcription and IFN-stimulated genes, thereby inhibiting the early inflammatory response induced by virus entry [50]. In addition, E1A protein interacts with MECL-1 to down-regulate the expression of MECL-1, thereby reducing antigen presentation on infected cells [51]. The E1A protein also inhibits the Rb pathway (Figure 5) [52]. Therefore, the versatility of the E1A protein plays an integral role in the HAdV immune escape and amplification [53]. (2) The E1B gene and E4 gene are involved in maintaining the survival of infected cells and promoting HAdV replication. The products of E1B-19K and E1B-55K jointly block E1A-induced apoptosis. E1B-55K also blocks IFN-stimulated gene expression and other innate immune responses. The E4 region encodes at least seven different proteins involved in late viral gene expression, DNA damage response, and apoptosis. E1B-55K and E3 ubiquitin ligase encoded by E4 open reading frame 6 (E4orf6) trigger proteasome-mediated degradation of the death domain-associated protein (Daxx) [54]. They jointly induce the selective export of late viral mRNA from the nucleus to the cytoplasm and inhibit the Rb pathway (Figure 5) [52,55]. E1B-55K could also inhibit the innate immune response by acting together with E4orf3 to prevent the production of IFN. Additionally, they reposition the Mrell-Rad50-Nbs1 complex, eliminating the formation of connexin and DNA damage signals during virus replication, which increases virus production from infected cells [56]. (3) E3 protein is the most famous immunomodulatory molecule. E3-gp19K prevents the transport of MHC-I to the plasma membrane, thereby reducing the invasion of leukocytes into infected cells. It also reduces the cell surface level of NK cell receptors, which further prolongs the survival of infected cells [57]. In addition, E3-6.7K, E3-10.4K, E3-14.5K, and E3-14.7K block exogenous apoptosis by down-regulating death receptors. (4) Non-coding RNA VA-I and VA-II also contribute to the survival and proliferation of HAdV by inhibiting the host immune system. Pol-III transcribes VA-I and VA-II genes into two short non-coding RNAs, respectively. In the transcription process of double-stranded DNA viruses, Pol-III transcribes 5′-triphosphorylated RNA, which is sensed by a DEAD-box helicase (RIG-I) and converted into an IFN-I response through the mitochondrial antiviral signal protein [58]. Furthermore, VA-I and VA-II not only suppresses IFN-induced PKR to alleviate the inhibition of protein synthesis in the antiviral state, but also binds to RIG-I to block the IFN-I response.

In general, after HAdV enters human body, the innate immune system monitors the virus through the activation of the systemic pro-inflammatory state and the enrichment of cytotoxic immune cells. After that, the immune system pushes the virus to accumulate through macrophages and triggers inflammation through cytoplasmic DNA sensing to identify and clear the virus. Under the promotion of the innate immune system, the adaptive immune system produces antibodies of multiple serotypes of HAdV to fight against its invasion. However, HAdV suppresses the immune response through early protein and non-coding RNA, thereby escaping the host’s clearance.

### 2.5. Advantages of Adenovirus as Oncolytic Virus

Compared with other viral vectors, oncolytic adenovirus has many unique advantages, including: (1) a wide range of host cells and low pathogenicity in humans; (2) the genetic structure and function of the virus have been thoroughly studied, and hence, it is easier to manipulate its genome; (3) the infection of adenovirus is not impacted by the cell cycle; (4) the virus can be effectively propagated, and it is convenient to prepare and purify to high infection titres; (5) with a broad infection spectrum, HAdV can infect many types of tumours, especially adenocarcinoma cells; (6) after infection, it will not be integrated into host chromosomes, which is helpful for reducing the risk of recombination mutations; (7) and it can express multiple genes simultaneously. Given that HAdV exhibits great advantages, it has become the most used oncolytic virus both as a bioactive therapy (lyses of cancer cells) or as a gene delivery vector in gene therapy research [59,60,61,62,63].

### 2.6. Genetic Modifications of Oncolytic Adenovirus

Most investigations on the effects of oncolytic adenovirus are still in the preclinical stages (Table 2 listed genetic modifications of oncolytic adenoviruses for studies in prostate cancer), where adenoviruses are genetically modified, either by improving tissue-specific targeting or enhancing tumour-specific killing effects while minimally damaging adjacent normal cells. The genetic modification of HAdV is effective in improving its accuracy of targeting tumour cells, improving its efficacy of the treatment, activating an anti-tumour immune response, and minimizing the damage to normal cells engendered by oncolytic viruses. The common strategies of oncolytic adenovirus modification include the deletion of viral genes to prevent its replication in normal cells, and the addition of immunoregulatory factors to enhance host immunity. After modification, the oncolytic adenovirus can better target the infected tumour cells and enhance the anti-tumour immune response of the body [64,65,66].

#### 2.6.1. Modification Based on Targeting Mechanisms

As expression levels of CAR on tumour cells are usually low, HAdV is more likely to infect normal cells [67]. However, in the tumour microenvironment, the function of CTL is often suppressed, and the secretion of IFN-γ is reduced, which is favourable for oncolytic adenovirus to replicate in tumour cells. This leads to a deficiency of a complete antiviral immune response in tumour cells, rendering tumour cells susceptible to infection by oncolytic viruses and making them replicate in a large quantity to exert oncolytic effects [68].

(1)Genetic modifications for specific infections

The structure of HAdV capsid protein is modified by gene editing to specifically recognize tumour cell surface receptors, thereby achieving the specific infection of tumour cells [69]. One strategy is to embed single-chain antibodies into HAdV fibre. With the assistance of antibodies, the viral fibre can bind to certain receptors (such as epidermal growth factor receptor, EGFR) highly expressed on the surface of tumour cells, resulting in selective infection [70]. Another strategy is to remodel HAdV fibre into a heterogeneous chimeric fibre, thereby enhancing the efficiency of virus infection of tumour cells and reducing toxicity, such as the oncolytic adenovirus AdΔ24RGD [71].

In addition, the extracellular matrix (ECM) in solid tumours can affect the infection and spread of oncolytic adenovirus. Matrix-degrading enzymes improve tumour permeability by degrading ECM, strengthening the spread of viruses, and increasing the concentration of viruses in tumour cells [72]. Therefore, oncolytic viruses expressing matrix-degrading enzymes can achieve better efficiency of infection.

(2)Genetic modifications for selective replication

To enable oncolytic adenovirus to replicate selectively in tumour cells, deactivation of some signalling pathways can improve the efficiency. One method is to delete the genes that are necessary for virus replication in normal cells, but not in tumour cells [73]. For example, viral infection leads to the activation of wild type p53, which restrains the replication of adenovirus, while viral E1B55K protein blocks the activity of p53 [74,75]. Deletion of the E1A CR2 region or the E1B55K gene in the viral genome will not permit virus replication in normal cells, while it can still replicate in tumour cells with the mutant and dysfunctional p53. Onyx-015 and H101 with E1B55kD gene deletion is this sort of genetically engineered virus.

Another commonly used strategy is to regulate the expression of genes necessary for HAdV replication. This is in order to place a promoter to permit the expression of essential genes for viral replication in tumour cells but not in normal cells such that the virus will only replicate in tumour cells [76]. Commonly used regulatory sequences include promoters of survivin, human telomerase reverse transcriptase (hTERT) and carcinoembryonic antigen (CEA), alpha-fetoprotein (AFP), and hypoxia response element (HRE). hTERT is highly expressed in tumour cells, but is low or not expressed in normal cells. Therefore, the hTERT gene promoter can be used to regulate the E1A gene of HAdV and confine virus replication in tumour cells [77].

Currently, increasing evidence indicates that microRNAs (miRNAs) play important roles in regulating virus–host cell interactions. On the one hand, RNA interference promotes the defence of viruses in many multicellular organisms [78,79]. On the other hand, some mammalian viruses were shown to benefit from the RNAi machinery of their host [80]. Interestingly, part of the host miRNAs can enhance virus replication. For example, the expression of miR-122, miR-501-3p, and miR-619-3p in liver cells promoted hepatitis C virus proliferation [81,82,83], and the elevated miR-132 expression in T cells augmented HIV-1 replication [84]. Therefore, using oncolytic adenovirus as a vector to deliver viral-promoted miRNA genes is a feasible way to increase the efficiency of oncolytic virus replication. miR-26b is a candidate that promotes the survival and proliferation of adenovirus. It has been reported that the overexpression of miR-26b inhibits the activation of adenovirus-induced NF-κβ, augments adenovirus-mediated cell death, increases the release of adenovirus progeny, and promotes adenovirus propagation and spread in human prostate cancer cell lines [78]. Exogenous expression of miR-26b may also contribute to cancer-selective replication of oncolytic adenovirus, since miR-26 is involved in blocking G1/S-phase progression by repressing CDK6 and cyclin E1, resulting in reduced phosphorylation of pRb. Therefore, combination of miR-26b with oncolytic adenovirus is potentially an effective therapeutic strategy for prostate cancer.

#### 2.6.2. Genetic Modifications with Anti-Tumour Molecules

By inserting a gene that encodes a product with tumour-killing activity into the HAdV genome, the virus directly serves as a delivering vector. Thus, the target gene is continuously expressed in tumour cells as the virus replicates, further enhancing the killing activity of the oncolytic adenovirus. For instance, (1) insertion of genes expressing immune-inflammatory mediators in the HAdV genome can increase the local immune-inflammatory mediators in the tumour microenvironment, thereby turning on the anti-tumour immune response and enhancing the anti-tumour efficacy of the oncolytic adenovirus. GM-CSF is currently the most generally used cytokine in clinical trials and has been inserted into the genomes of diverse oncolytic viruses. Its roles in improving therapeutic effects have been shown in several preclinical trials [85,86,87,88]. Additionally, other cytokines such as interleukin-2 (IL-2), IL-24, IFN, tumour necrosis factor (TNF) α, and soluble CD80 and CCL3 have been used in the study of viral treatment of tumours [89,90,91]. When cytokines and chemokines reach the tumour microenvironment, they can override the suppressed state of immune cells and yield a therapeutic immune response; (2) a prodrug-converting enzyme converts non-toxic substrates into biologically active drugs, such that the drugs exert a specific killing effect in tumour cells infected with oncolytic adenoviruses [92]. Carboxypeptidase G2 (CPG2) is an effective prodrug-converting enzyme [93]. When the CPG2 recombinant adenovirus is used in combination with the prodrug ZD2767P, it significantly increases the sensitivity of tumour cells to the latter; (3) anti-tumour angiogenesis genes could also be carried by the oncolytic virus to inhibit tumour progression. The formation of new blood vessels is an essential step for tumours to expand. The oncolytic adenovirus expressing tumour metastasis suppressor protein (KISS-1) not only enhances the toxicity of the virus to tumour cells, but also inhibits the brain metastasis of tumour in the mice models of breast cancer by inhibiting the activity of VEGF and MMP-14 required for angiogenesis, thereby inhibiting the distant metastasis of tumour cells [94]; (4) apoptin is a tumour-specific pro-apoptotic protein which plays a vital anti-tumoral role in a variety of cancers, including melanoma, lymphoma, colon carcinoma, and lung cancer [95,96,97,98]. It senses an early event of oncogenic transformation, promoting cancer-specific apoptosis, without hurting normal cells. Therefore, apoptin is a potential future anticancer therapeutic agent [99]. Cui et al. have created a dual cancer-specific oncolytic adenovirus Ad-Apoptin-hTERTp-E1 expressing both apoptin and E1, which allows the replication of adenovirus in tumour cells, and enables the expression of the apoptin protein, thereby effectively killing the tumour [100]. This effect has been validated in prostate cancer PC-3 cells [99].

#### 2.6.3. Using Carriers

To enhance the efficiency of the oncolytic adenovirus to reach the site of tumour, carriers are used as the “Noah’s Ark” that effectively escapes the immune surveillance system [101]. Given that immune cells can infiltrate the tumour microenvironment and avoid immune surveillance, they can be used as cell carriers to transport oncolytic viruses. The cell carriers under study include tumour-infiltrating lymphocytes (TIL) [102], mesenchymal stromal cells (MSCs) [103,104], and cytokine-induced killer cells [105,106]. MSCs can be obtained from patients relatively easily, rapidly expanded in vitro and easily modified. It is crucial that MSCs also have tumour tropism and can be used as an ideal carrier to deliver biological anti-tumour agents. Studies have shown that MSCs as carriers can deliver HAdV to tumour to exert significant anti-tumour effects [107].

In addition to immune cells, CCL2-coated liposomes could also be used as virus carriers to recruit CCR2-expressing circulating monocytes into the tumour microenvironment [108]. Since liposomes are composed of naturally occurring phospholipid, their biological characteristics are unreactive and indistinguishably immunogenic. Moreover, liposomes can be used to carry both hydrophilic and hydrophobic drugs; given that liposomes were created by mimicking the cell membrane, they contain an aqueous core to encapsulate hydrophilic solutes, and can be easily internalized by cells. Hydrophobic substances could be entrapped in the lipid bilayer of liposomes for transportation. The roles of oncolytic adenovirus Ad[I/PPT-E1A] carried by CCL2-coated liposomes were identified in both in vitro experiments and animal models [108], which was considered a promising strategy.

In recent years, cationic nanoparticles have attracted the attention of researchers. Through electrostatic interactions, positively charged gold nanoparticles (AuNP) could coat the highly negatively charged viral particle, which facilitates the attachment of the virus to negatively charged cell membranes by bringing the viral fibre and penton proteins closer to the cellular CAR and integrin receptors. Then, the viral infection and propagation processes will be more efficient in cancer cells [109]. Man et al. used AuNP complexing with Ad∆∆ and Ad-3∆-A20T; then, the replication and cell-killing abilities of oncolytic adenovirus were significantly enhanced. Compared with cell therapy, AuNP-coated oncolytic adenovirus is smaller, easier to produce, and more versatile, which can penetrate and spread within the tumour microenvironment. Although the uptake of oncolytic virus is also increased in healthy cells, due to the gene deletions of Ad∆∆ and Ad-3∆-A20T, their replication and spread will not proceed to influence the function of normal cells.

### 2.7. The Mechanism Underlying the Action of Oncolytic Adenovirus

The key to tumour treatment with oncolytic virus is the specific replication of the viruses in tumour cells rather than in normal cells. The direct consequence of virus replication is the lysis and death of tumour cells. Then, newly produced and released progeny of oncolytic virus continue to infect neighbouring tumour cells and achieve repeated killing [110]. More importantly, the tumour antigen released during oncolysis also stimulates specific anti-tumour immunity to further improve the therapeutic effect.

#### 2.7.1. Virus-Mediated Tumour-Killing Mechanism

Oncolytic adenoviruses generally exert anti-tumour effects by directly killing tumour cells, disrupting tumour vasculature and/or carrying suicide genes. The specific mechanisms are summarized as follows (Figure 6) [111]: (1) oncolysis occurs usually due to changes in signal transduction pathways in tumour cells, such as the disruption of membrane-associated tyrosine protein kinase signal transduction events linked to transcriptional regulation [112]. First, when adenovirus enters host tumour cells, cell cycle progression slows down to make the cells ready to amplify the virus, and second, a large number of progeny viruses are released, followed by cell lysis. Through the “bystander effect”, uninfected tumour cells are subsequently killed [113]. Finally, cell death ensues in the form of apoptosis, necrosis, or autophagic cell death. In addition, HAdV produces substances with cytotoxicity and oncolytic activity. For example, the death protein encoded by the E3 region directly mediates tumour cell killing [114]. (2) Oncolytic adenovirus induces apoptosis or necrosis of uninfected tumour cells through anti-angiogenesis and anti-vasculature effects [115,116]. (3) Oncolytic adenoviruses can also carry suicide genes to enhance its tumour-killing effect. Thymidine kinase (HSV-TK) combined with the non-toxic gancyclovir (GCV) is an enzyme prodrug system commonly used in tumour suicide gene therapy [117]. GCV phosphorylated by the TK gene interferes with the DNA synthesis of tumour cells and promotes cell apoptosis.

#### 2.7.2. The Anti-Tumour Immune Response Mechanism

The interaction between virus and host tumour cells is complex [118]. On the one hand, the antiviral immune response is the main restrictive factor for virus amplification, which weakens the anti-tumour effect of the virus. On the other hand, the killing effect of the immunity in virus-infected tumour cells is enhanced. The infection of tumour cells by the virus induces the infiltration of lymphocytes and antigen-presenting cells (APC) to the infection site. Virus-infected tumour cells are recognized and killed by cytotoxic lymphocytes due to the presentation of viral antigens. After that, the tumour antigens released after lysis of tumour cells enhance the antigen presentation ability of APCs, thereby generating a specific immune response and finally forming a long-term anti-tumour immune response.

Oncolytic adenovirus induces immune memory, which is essential to prevent tumour recurrence and metastasis [76]. Although tumour is present in an immunosuppressive microenvironment, the oncolytic adenovirus stimulates a robust immune response, and recruits and activates TIL in the tumour microenvironment to exert antiviral response, which helps to break or even reverse the tumour’s immune tolerance state [119]. More importantly, oncolytic adenovirus further presents tumour-specific antigens to CD8+T cells, forming a potentially effective anti-tumour immunity. Oncolytic adenovirus has also been demonstrated to induce immune cells to produce IFN-I and express immune costimulatory molecules and activate the antigen presentation signal pathway [120].

## 3. Oncolytic Adenovirus in Clinical Trials

The development of adenovirus as an oncolytic virus is relatively easy as the technique is mature and available in many laboratories. Table 3 shows the currently ongoing clinical trials using oncolytic adenovirus in the treatment of prostate cancer. In addition, the completed clinical trials are described as follows:

### 3.1. CV706 (CN706)

CV706 (CN706) was the first reported oncolytic adenovirus in the treatment of prostate cancer. In 20 cases of prostate cancer recurring after radiotherapy, the virus was directly injected into the prostate using transrectal ultrasound positioning via the perineum [14]. The results of phase I clinical trial showed its safety and certain efficacy. The liver function test showed a certain degree of antiviral activity, and the PSA level was reduced.

### 3.2. CG7870 (CV787)

The oncolytic adenovirus GG7870 (CV787) was constructed by Yu et al. and used in phase I clinical trial in the treatment of prostate cancer [15]. Intravenous injection of the virus to patients with castration-resistant prostate cancer showed that 70% (16/23) of the patients had the virus in the blood. The adverse reactions were flu-like symptoms (such as fever, fatigue, chills, nausea, and/or vomiting). However, the therapeutic effect was not satisfactory. Only 5 of the 23 patients had a 25–49% decrease in serum PSA. When CG7870 was used in combination with docetaxel, 36% of patients had a decrease in PSA levels, and 27% of patients showed no tumour progression within 6 months. However, subsequent research did not produce better results [16].

### 3.3. Ad5-CD/TKrep (FGR)

Human species C adenovirus serotype 5 (Ad5) is the archetype oncolytic adenovirus that has been used in several investigations. The oncolytic adenovirus Ad5-CD/TKrep (FGR) that expresses a suicide fusion gene to treat prostate cancer [17] is constructed by deletion of E1B55K and E3 and fusion of cytosine deaminase (CD) and herpes simplex virus type-1 thymidine kinase (HSV-1 TK). CD and HSV-1 TK render prostate cancer sensitive to 5-fluorocytosine (5-FC) and GCV. The virus was injected under the rectal ultrasound guidance into 16 prostate cancer patients who had recurred cancer after radiotherapy [18]. Two days later, the patients were given 5-FC/GC for treatment. The results showed that 94% of patients had mild to moderate adverse reactions, but all had different degrees of remission; 44% (7/16) of patients had serum PSA decreased by more than 25%; 19% (3/16) of patients had serum PSA was less than 50% of the level prior to treatment. The tumour histology after treatment showed transgene expression and tumour necrosis. Follow up study of these 16 patients for 5 years reported that 88% (14/16) patients survived [19]. Another study using Ad5-CD/TKrep intra-prostatic injection combined with external radiotherapy to treat 15 newly diagnosed middle- and high-risk prostate cancer patients followed up an average 9 months [20]. The results showed that the combination therapy was significantly better than the results of single radiotherapy. Therefore, the results suggest that for patients with PSA recurrence after radiotherapy, treatment with oncolytic adenovirus Ad5-CD/TKrep that expresses the suicide gene is a good choice.

### 3.4. Ad5-yCD/mutTKSR39rep-ADP

Freytag et al. have constructed additional suicide genes (improved yeast CD and mutant SR39 HSV-1 TK) into oncolytic adenovirus, namely Ad5-yCD/mutTKSR39rep-ADP and combined this virus with intensity-modulated radiotherapy (IMRT) in phase I prostate cancer clinical trial [21]. Their results revealed safety and sound therapeutic effects. Later on, they reported on a prospective phase II clinical study of Ad5-yCD/mutTKSR39rep-ADP combined with IMRT in the treatment of intermediate-risk prostate cancer [22]. The results showed that the biopsy positive rate of the gene therapy combined with IMRT treatment group decreased by 42%, as compared with IMRT alone group. There was no significant difference in the quality of life between the combination treatment and the single treatment group, and 84% of patients had a needle biopsy within 2 years of treatment. Additionally, the combination therapy did not aggravate the patients’ adverse reactions but reduced the positive rate of biopsy after treatment.

## 4. Existing Limitations and Future Directions

In recent years, using oncolytic adenovirus for prostate cancer treatment has made encouraging progress. Preclinical and clinical studies have demonstrated the antitumour effect of oncolytic adenovirus in prostate cancer and applicability in safety and reliability. It appears to be in a leading position in the field of gene therapy. Especially, the combination of oncolytic adenovirus and other antitumour strategies would create a new era in cancer therapy.

Although the oncolytic adenovirus has achieved significant effects in both preclinical and clinical settings, there is still much space for improvement. With continuous understanding of oncolytic adenovirus, it can be optimized through genetic manipulation, cytokine loading, carrier packaging and other means to make it more effective for cancer therapy. However, due to the presence of viral immunogenicity and issues of tissue specific targeting without damage to normal tissues, most oncolytic adenoviruses are still in the preclinical and early clinical trial stages. However, issues that limit its utility in cancer therapy still need to be considered. For example, oncolytic adenoviruses have complex biological characteristics, including host specificity and virus–host interaction; These biological characteristics remain not be fully understood in humans. Moreover, the inherent immunogenicity of oncolytic adenovirus can easily cause local tissue inflammation and possible overt immune response of the whole body.

Due to the high level of its receptor CAR expressed on the surface of hepatocytes, adenovirus is highly hepatotropic, thereby increasing liver toxicity [126]. Furthermore, after entering the blood circulation, most of the virus are cleared by Kupffer cells, which significantly reduces the systemic effect on targeted tissue [127]. The complex system composed of oncolytic adenovirus, tumour cell microenvironment and the immune system plays integral roles in the efficacy of oncolytic adenovirus therapy. Additionally, more than 50 serotypes of adenovirus make it difficult to choose the type of HAdV specific to individual tumour tissues and require a lot of in-depth and detailed research [128]. As adenovirus is a common pathogen in humans, the pre-existing immunity may quickly clear the oncolytic adenovirus after injection [129]: (1) The treatment-induced immune response would inhibit the infection and transmission of the virus. (2) The natural antiviral immune response limits viral replication and amplification. (3) The infiltration of immune cells and neutralizing antibodies prevent the spread of the virus. Therefore, oncolytic adenovirus can only be used to treat tumour by intra-tumoral injection, which significantly limits its clinical application.

Of note, natural occurring tumour is a complex and with the heterogeneous identity. The preclinical data are obtained with xenograft tumour models derived from single cell lines in SCID mice that are in immunodeficient animals. However, there are considerable differences between experimental animals and humans in terms of their microenvironment and the biological characteristics of tissue cells.

Oncolytic adenovirus targeted therapy is currently more suitable for local treatment. However, the systemic therapy still needs to overcome the following problems [1,130]: (1) How does oncolytic adenovirus overcome the immune surveillance by neutralizing antibodies in circulation and successfully reaching the lesion? (2) How does the oncolytic adenovirus pass through the barrier of blood vessel walls to avoid transcytosis of endothelial cells and then travel to target cells? (3) Tumour cells are always surrounded by membranous structures formed by stromal cells. The oncolytic adenovirus needs to pass through the matrix and membrane structure to reach tumour site, which dramatically reduces the efficiency of the virus to spread in tumour.

## 5. Conclusions

In summary, after continuous exploration and research, the mechanism of oncolytic adenovirus therapy to kill tumours has become increasingly apparent, and clinical trials for the treatment of prostate cancer are gradually being launched. As a novel tumour therapeutic method, it complements the shortcomings of traditional treatment including poor specificity, intolerable side effects, and cross resistance. The combined use of oncolytic adenovirus and traditional therapeutic methods has exhibited synergistic effects and excellent application prospects. However, in the process of oncolytic adenovirus application in clinic, there are still pitfalls that need to be overcome. With the continuous development of biological technology, the safety, specificity, and oncolytic properties of oncolytic adenovirus in cancer targeted therapy could be better improved. More oncolytic adenoviruses are waiting to enter clinical trials and to be truly applied in clinic for the benefit of prostate cancer patients.

## Figures and Tables

**Figure 1 biomedicines-10-03262-f001:**
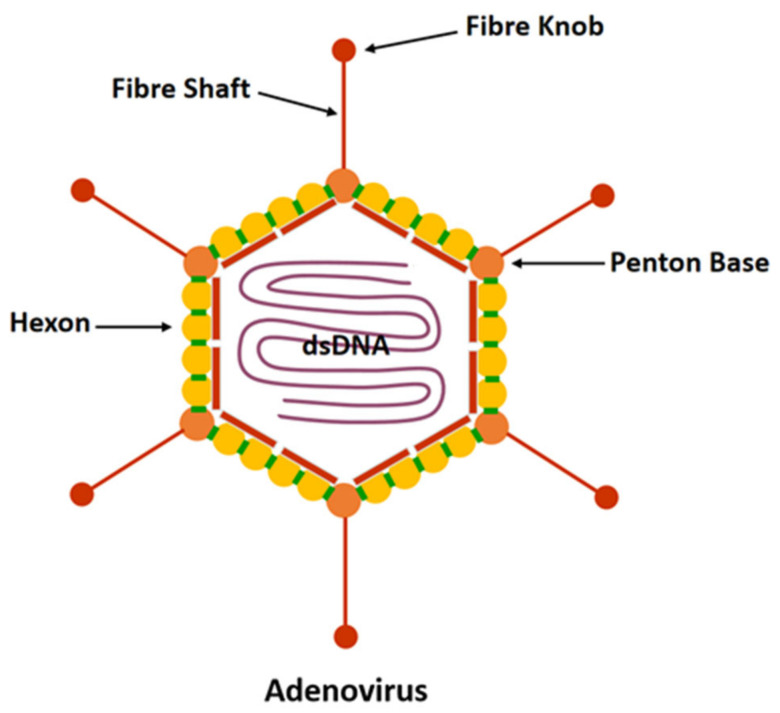
HAdV structure.

**Figure 2 biomedicines-10-03262-f002:**
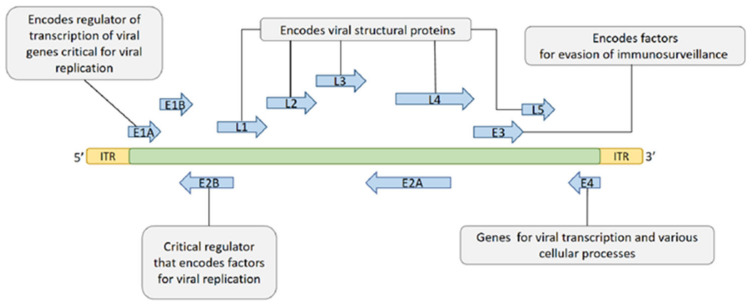
The genes and functions of HAdV.

**Figure 3 biomedicines-10-03262-f003:**
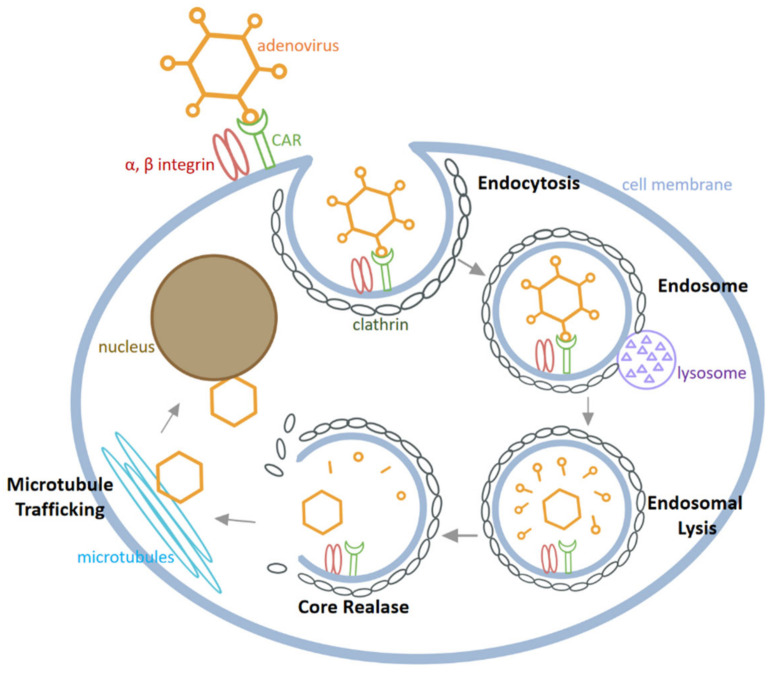
The process of HAdV infecting cells.

**Figure 4 biomedicines-10-03262-f004:**
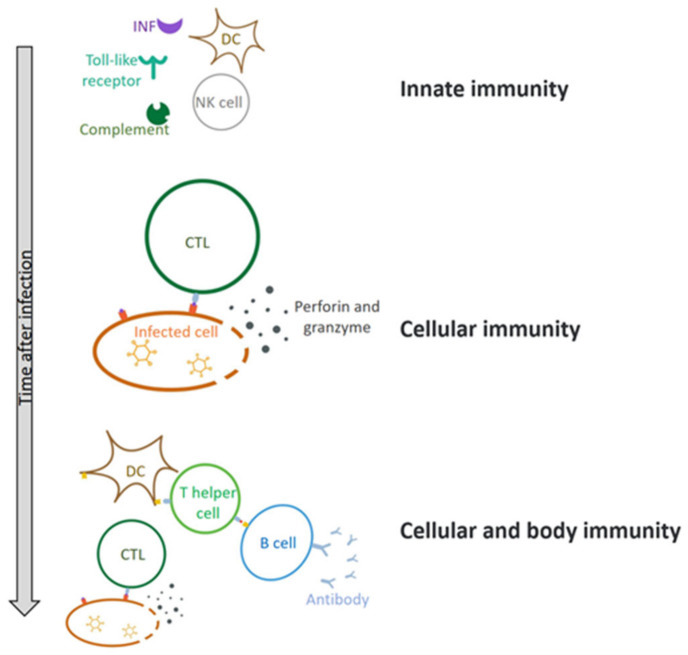
The immune response of HAdV after it enters the body. The body first produces an innate immune response, involving dendritic cells (DC cells), natural killer cells (NK cells), interferon (INF), and so on. Over time, cellular and humoral immunity comes into play. CTL—Cytotoxic T lymphocyte.

**Figure 5 biomedicines-10-03262-f005:**
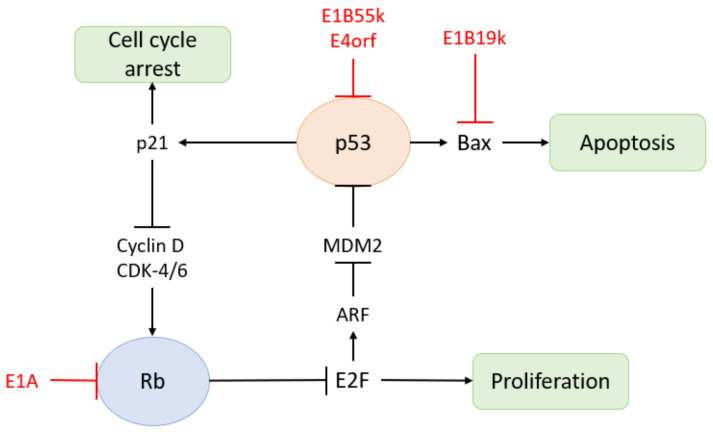
Proteins of HAdV can inhibit Rb pathway and p53 pathway.

**Figure 6 biomedicines-10-03262-f006:**
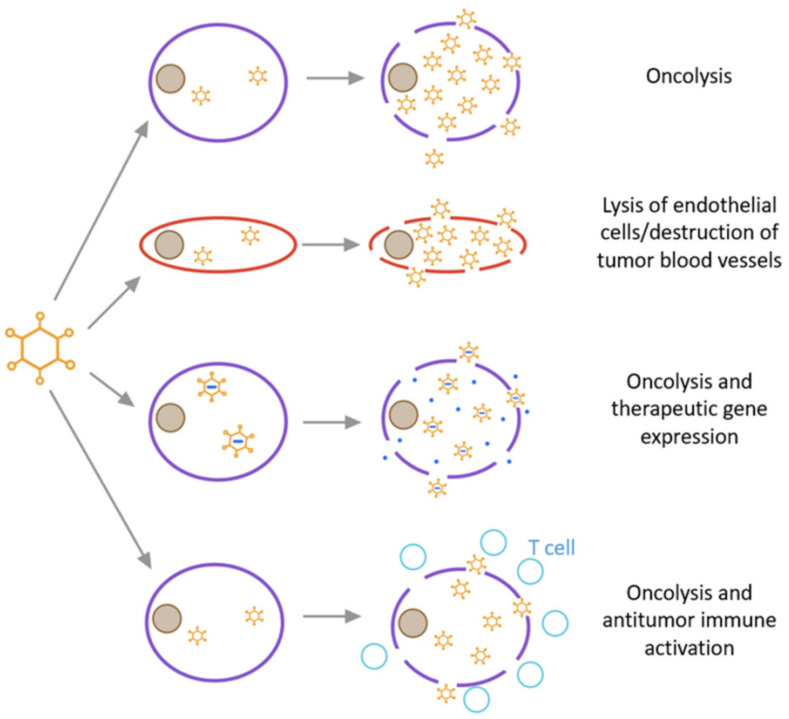
Types of oncolytic viruses and their mode of action.

**Table 1 biomedicines-10-03262-t001:** Classification of HAdV and its receptors (information summarized from Arnberg et al. (2012) [39] and Gao et al. (2020) [40]).

Subclasses	Serotypes	Identified Receptor(s)	Major Site(s) of Infection
A	12, 18, 31, 61	CAR	Cryptic (GI tract, respiratory tract, urinary tract)
B	3, 7, 11, 14, 16, 21, 34, 35, 50, 55, 66, 68, 76, 77, 78, 79	CD46, CD68, CD80, DSG2	Respiratory tract, eye, urinary tract, GI tract
C	1, 2, 5, 6, 57, 89	CAR, HSPG, MHC1-a2, SR, VCAM-1	Respiratory tract, eye, lymph, liver, urinary tract, GI tract
D	8, 9, 10, 13, 15, 17, 19, 20, 22-30, 32, 33, 36, 37, 38, 39, 42–49, 51, 53, 54, 56, 58, 59, 60, 62-65, 67, 69–75, 80–88, 90–103	CAR, CD46, SA	Eye, GI tract
E	4	CAR	Respiratory tract, eye
F	40, 41	CAR	GI tract
G	52	CAR, SA	GI tract

**Table 2 biomedicines-10-03262-t002:** Modification strategies and preclinical experiments of some oncolytic adenoviruses.

Modifications Strategies	Effects	Name	Modification Details	Results
Modifications at the transduction level	Enhance the infection efficiency of HAdV	Ad5-D24RGD	Link Arg-Gly-Asp (RGD) peptide chains in the fibre portion of HAdV and mutation of the Rb binding site of E1A	Can specifically bind to the integrin αvβ3 or αvβ5 on the cell surface and can specifically replicate in Rb-mutated cells
		AxdAdB3/F-RGD	Link Arg-Gly-Asp (RGD) peptide chains in the fibre portion of HAdV and mutation of E1A and E1B	Can specifically bind to the integrin αvβ3 or αvβ5 on the cell surface and can replicate in Rb-mutated cells and p53-mutated cells
		Ad.5/3-CTV	Replace Ad5’s fibre knob with Ad3’s fibre knob	Can specifically target the Ad3 receptor CD46
Modifications at the cell cycle-dependent replication selectivity	Enhance the replication selectivity of HAdV	ONYX-015 (dl1520)	Deletion of E1B55KD	Can specifically replicate in p53-mutated cells
		dl922-947 (AxE1AdB) and Δ24	Mutation of Rb binding site of E1A	Can specifically replicate in Rb-mutated cells
		AxdAdB-3	Mutation of E1A and E1B	Can replicate in Rb-mutated and p53-mutated cells
		AdΔΔ	Mutation of Rb binding site in E1A region and deletion of E1B19kD and retains E3 region	Can specifically replicate in Rb-mutated cells and can sensitize apoptosis of normal cells and can suppress host immune response
Modifications at the tissue-specific promoter-regulated replication selectivity	Enhance the replication selectivity of HAdV	CN706 (CV706, Ad-PSEE1a)	Insert PSA promoter/enhancer in E1A region	Can selectively replicate in PSA-producing cells
		CV764	Insert PSA enhancer (PSE) in E1A region and insert hK2 enhancer/promoter in E1B region	Can selectively replicate in PSA-producing cells
		CV787	Insert probasin promoter in E1A region and insert PSE in E1B region and retains E3 region	Can selectively replicate in PSA- and probasin-producing cells and can suppress host immune response
		Ad5PB-RSV-NIS	Insert probasin promoter in E1A region and replace E3 with NIS gene	Can selectively replicate in probasin-producing cells and can express NIS
		Ad.PSMApro-hNIS	Insert PSMA promoter and Insert NIS gene	Can selectively replicate in PSMA-producing cells and can express NIS
		Ad-PSMA(E-P)-CD	Insert PSMA enhancers and promoters	Can selectively replicate in PSMA-producing cells
		Ad-PSES-luc	Insert PSE and PSMA enhancer	Can selectively replicate in PSA- and PSMA-producing cells
		Ad[I/PPT-Luc] and Ad[I/PPT-E1A]	Insert PSE, PSMA enhancer and TARP promoter	Can selectively replicate in PSA-, PSMA-, and TARP-producing cells
		Ad.DD3-E1A-IL-24 and Ad-DD3p-E1A	Insert DD3PC3 promoter	Can selectively replicate in DD3PC3-producing cells
		Ad-OC-E1a	Insert OC promoter in E1A region	Can selectively replicate in OC-producing cells
		OBP-301	Insert hTERT promoter in E1A region	Can selectively replicate in hTERT-producing cells
		Ad-BSP-Ela	Insert BSP promoter in E1A region	Can selectively replicate in BSP-producing cells
Modifications with multiple targeting strategies	Enhance the infection efficiency and replication selectivity of HAdV	Ad5/35PSES.mRFP/ttk	Replace Ad5’s fibre knob with Ad35’s fibre knob and insert PSE and PSMA enhancer and insert mRFP/ttk fusion protein	Can specifically target the CD46 and can selectively replicate in PSA- and PSMA-producing cells and can express mRFP/ttk
		ORCA-010 (Ad5-D24RGDT1)	Link RGD peptide chains in the fibre portion and Mutation of Rb binding site of E1A and mutation of E3/19K gene	Can specifically bind to the integrin αvβ3 or αvβ5 and can specifically replicate in Rb-mutated cells and can promote host cells lysis and viruses release
Oncolytic adenovirus as vector-mediated gene therapy	Make HAdV have the dual anti-tumour effect of oncolysis and gene therapy	Ad.sTβRFc	Insert sTGFbRIIFc	Can express sTGFbRIIFc and inhibit the TGF-β pathway
		Ad5-Δ24-sOPG-Fc-RGD	Link RGD peptide chains in the fibre portion and mutation of Rb binding site of E1A and insert sOPG and Fc	Can specifically bind to the integrin αvβ3 or αvβ5 and can specifically replicate in Rb-mutated cells and can express sOPG and Fc
		Ad5-PSE/PBN-E1A-ARC685Y	Insert PSE and insert mutated androgen receptor (AR) cDNA in E1A region	Can selectively replicate in PSA-producing cells and can replicate in cells with both high and low androgen levels
		Ad-PL-PPT-E1A	Insert PSE, PSMA enhancer and TARP promoter and insert PSA-IZ-CD40L fusion gene	Can selectively replicate in PSA-, PSMA-, and TARP-producing cells and can activate the body’s anti-tumour immune response
Fusion of genes from different oncolytic adenoviruses	By combining the gene fragments of different HAdV to improve the therapeutic effect and reduce the side effects	Ad657	Use the HVR region of Ad57 to replace this region of Ad6 by homologous recombination	Compared with Ad5 and Ad6, Ad657 has similar in vitro oncolytic activity. Ad657 is associated with the lowest hepatotoxicity.
		RCAd11pADP	ADP gene, located from 29,491 to 29,772 nt in the Ad5 E3 region, was cloned into the Ad11pe1 shuttle vector; RCAd11p vector carrying ADP was constructed by inserting the gene into the Ad11p E1 shuttle vector at 451 nt upstream of the E1A region	Can significantly promote tumour apoptosis and effectively prolong the period of mice survival

**Table 3 biomedicines-10-03262-t003:** Some currently ongoing clinical trials of oncolytic adenovirus for the treatment of prostate cancer [121,122,123,124,125].

Study Title	Official Title	Intervention	Study Description	Phase	Status	Completion Date
Use of Recombinant Adenovirus Therapy to Treat Localized Prostate Cancer (NCT01931046)	A Phase 1/2a Study of In-situ REIC/Dkk-3 Therapy in Patients With Localized Prostate Cancer (MTG-REIC-PC003)	Drug: Ad5-SGE-REIC/Dkk3	The purpose of this study is to evaluate the safety and effectiveness of AD5-SGE-REIC/Dkk-3 in patients with localized prostate cancer.	Phase 1 Phase 2	Completed, no results	March 2020
A Phase I/II, Safety Clinical Trial of DCVAC/PCa and ONCOS-102 in Men With Metastatic Castration-resistant Prostate Cancer (NCT03514836)	A Phase I/II, Clinical Trial to Evaluate the Safety and Immune Activation of the Combination of DCVAC/PCa, and ONCOS-102, in Men With Advanced Metastatic Castration-resistant Prostate Cancer.	Biological: DCVac/PCa (ONCOS-102) Drug: Cyclophosphamide	This open label, dose escalating study is a phase I/IIa first in man study designed to evaluate the safety and tolerability of intratumoral administration of a novel oncolytic adenovirus (ORCA-010) in treating diagnosed treatment naïve Patients with localized prostate cancer.	Phase 1 Phase 2	Terminated (Insufficient Accrual)	25 January 2021
A Clinical Trial of AdNRGM Plus CB1954 in Prostate Cancer (AdUP) (NCT04374240)	Phase I Trial of Replicative Defective Type 5 Adenovirus Vector Expressing Nitroreductase & GMCSF Given Via Trans-perineal Template-guided Intra-prostatic Injection Followed by iv CB1954 in Locally Relapsed Prostate Cancer Patients	Genetic: AdNRGM	This is an open label, non-randomised, phase I, sequential group trial which will explore the safety and tolerability of ascending doses of AdNRGM, in combination with CB1954.	Phase 1	Completed, no results	November 2021
Phase 1 Trial of Interleukin 12 Gene Therapy for Locally Recurrent Prostate Cancer (NCT02555397)	Phase 1 Trial of Oncolytic Adenovirus-Mediated Cytotoxic and Interleukin 12 Gene Therapy for Locally Recurrent Prostate Cancer After Definitive Radiotherapy.	Biological: Ad5-yCD/mutTKSR39rep-hIL12	The primary purpose of this phase 1 study is to determine the dose-dependent toxicity and maximum tolerated dose (MTD) of oncolytic adenovirus-mediated cytotoxic and IL-12 gene therapy in men with locally recurrent prostate cancer after definitive radiotherapy.	Phase 1	Active, not recruiting	14 February 2023
First in Man Clinical Study to Evaluate Safety and Tolerability of an Oncolytic Adenovirus in Prostate Cancer Patients (NCT04097002)	A Phase I/IIa Study Evaluating the Safety and Tolerability of Intratumoral Administration of ORCA-010 in Treatment-Naïve Patients With Localized Prostate Cancer.	Biological: ORCA-010	This open label, dose escalating study is a phase I/IIa first in man study designed to evaluate the safety and tolerability of intratumoral administration of a novel oncolytic adenovirus (ORCA-010) in treating diagnosed treatment naïve Patients with localized prostate cancer.	Phase 1 Phase 2	Recruiting	December 2023

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
