# Peer review of "Oncolytic Adenovirus, a New Treatment Strategy for Prostate Cancer"

_biomedicines, 2022, doi:10.3390/biomedicines10123262_

Round 1

Reviewer 1 Report

This is a well written review of Oncoyitic Adenovirus for treating PCa. I congratulate authored for the extensive and properly presentation.

My comments and suggestions:

ABSTRACT

Line 13. Introduce %

INTRODUCTION

Line 57. Change sentence "RP is the most effective treatment...". It is not true¡. There is not robust evidence that CX is better than Radiotherapy in localized PCa. I would sugglest..."It is one of the most effective..."

Line 64. Complete sentence. "Docetaxel-based QT can increase survival...". Introduce also, Radiopharmaceutical treatments 8Ra-223, 177-Lu-PSMA) and references.

Author Response

We thank the reviewer 1 for the positive comments and constructive suggestions.  We made changes and hereby address the critiques point to point as follows.

ABSTRACT

  1. Line 13. Introduce %

Response:  Thanks for this suggestion. we searched literature and changed to “10% to 50% of cases are estimated to progress to metastatic castration-resistant prostate cancer (mCRPC) within 3 years of diagnosis, according to references (Nat Clin Pract Urol 6:76-85, 2009,  Lancet Oncol 16:e279-e292, 2015,PMID: 30260754)

INTRODUCTION

  1. Line 57. Change sentence "RP is the most effective treatment...". It is not true¡. There is not robust evidence that CX is better than Radiotherapy in localized PCa. I would suggest..."It is one of the most effective..."

Response: Thanks and changed.

  1. Line 64. Complete sentence. "Docetaxel-based QT can increase survival...". Introduce also, Radiopharmaceutical treatments 8Ra-223, 177-Lu-PSMA) and references.

Response: Thanks for the suggestion.  We made changes in Line 65.

Reviewer 2 Report

Comments for the review titled: “Oncolytic adenovirus, a new treatment strategy for prostate cancer

Line 38: Rewrite the sentence. It should read: “Currently, plenty of viruses have been used for oncolytic therapy, including adenovirus, herpes simplex…”

Line 40: It should be viruses instead of virus.

Line 53: Add the references for the clinical trials.

Line 64: Check whether Ref #18 is the appropriate one for this statement.

Figure 1: Since the image on the figure was adapted from Ref #25, the authors should include a statement to address this. Maybe they should include: Adapted from Stasiak and Stehle (2020).

Line 97: Since Table 1 is very similar to the tables on Refs #27 and #28, the authors should state that it was its prepared by combining information of the two studies. This could be added on the text citing the references or in a footnote under the table.

Line 104: Check whether these two references should be the same since they refer to the figure that was adapted from Ref #30.

Line 109: This reference does not seem to describe what is included on the text. Please check.

Figure 3: Maybe include a legend for some of the symbols that are not identified in the figure.

Line 384:  It is not clear whether the clinical trials described on the text are the same ones included on the Table 3.

Oncolytic adenovirus in clinical trials section: The authors should include for each clinical trial (if available): the Phase in which they are, whether the studies were completed, side effects, and any follow up on patients, if it was or is being done.

Table 3: Include references to the clinical trials; it could be the webpages at clinicaltrials.gov. If additional information is found regarding the current study results, please add.

Author Response

We thank this reviewer for careful reading our manuscript and giving the thoughtful comments and constructive suggestions.  Accordingly,we have made changes and hereby address them point to point as follows.

  1. Line 38: Rewrite the sentence. It should read: “Currently, plenty of viruses have been used for oncolytic therapy, including adenovirus, herpes simplex…”

Response: Thanks for the suggestion and changed.

  1. Line 40: It should be viruses instead of virus.

Response: Thanks for the suggestion and changed.

  1. Line 53: Add the references for the clinical trials.

Response: Thanks for the suggestion and added.

  1. Line 64: Check whether Ref #18 is the appropriate one for this statement.

Response: Thanks for pointing out the error and corrected.

  1. Figure 1: Since the image on the figure was adapted from Ref #25, the authors should include a statement to address this. Maybe they should include: Adapted from Stasiak and Stehle (2020).

Response: That’s a very good point and we added.

  1. Line 97: Since Table 1 is very similar to the tables on Refs #27 and #28, the authors should state that it was prepared by combining information of the two studies. This could be added on the text citing the references or in a footnote under the table.

Response: Thanks for the suggestion.  We made changes.

  1. Line 104: Check whether these two references should be the same since they refer to the figure that was adapted from Ref #30.

Response: Thanks for the suggestion.  We revised it, which is now in Line 114: ...... and 5 late transcription units (L1-L5) encoding the viral structural protein (Figure 2)40,41.

Figure 2. The genes and functions of HAdV40,41. (Adapted from Afkhami et al. (2016).)

  1. Line 109: This reference does not seem to describe what is included on the text. Please check.

Response: Thanks for pointing out the error.  It is corrected now.

  1. Figure 3: Maybe include a legend for some of the symbols that are not identified in the figure.

Response: Thanks for the suggestion. We redrew as the following by adding terminological words to the figure.

Figure 3. The process of HAdV infecting cells43. (Adapted from Bil-Lula et al. (2012).)

  1. Line 384: It is not clear whether the clinical trials described on the text are the same ones included on the Table 3.

Response:  Different, clinical trials finished were described in the text, while ongoing clinical trials listed in the Table 3.  This is noted in the revised version (from Line 399).

  1. Oncolytic adenovirus in clinical trials section: The authors should include for each clinical trial (if available): the Phase in which they are, whether the studies were completed, side effects, and any follow up on patients, if it was or is being done.

Response: Thanks for the suggestion. See Response 10

  1. Table 3: Include references to the clinical trials; it could be the webpages at clinicaltrials.gov. If additional information is found regarding the current study results, please add.

Response: Thanks for the suggestion. We added them.

Reviewer 3 Report

The paper summarizes the advantages and pitfalls of HAdV-based cancer therapy, as well as the strategies of its modification. Then, it discusses the results of recent  preclinical and clinical investigations on HAdV therapy and evaluates the potential of oncolytic adenovirus in prostate cancer treatment. The paper is well organized and written. It contains comprehensive information on the topic that that may be of interest to the broad readership of Biomedicines.

Author Response

We thank the reviewer 3 for approving our manuscript in the first round.
